# Evaluation of Probiotic Strains Isolated from *Artemisia argyi* Fermentation Liquor and the Antagonistic Effect of *Lactiplantibacillus plantarum* against Pathogens

**Hui Zhan [1,†], Yao He [1,†], Qi'an Wang [1], Qingzi Lu [1], Lihua He [1], Xueying Tao [1,2,*] and Hua Wei [1,2,3,*]**

[1] State Key Laboratory of Food Science and Resources, Nanchang University, Nanchang 330047, China; saibai0@126.com (H.Z.); hy18779168236@163.com (Y.H.); wqa121314@163.com (Q.W.); nculuqingzi@163.com (Q.L.); 13970902378@163.com (L.H.)

[2] International Institute of Food Innovation, Nanchang University, Nanchang 330299, China

[3] Jiangxi-OAI Joint Research Institute, Nanchang University, Nanchang 330047, China

\* Correspondence: taoxueying@ncu.edu.cn (X.T.); weihua@ncu.edu.cn (H.W.)

† These authors contributed equally to this work.

**Abstract:** This study was aimed at screening potential probiotic candidates to enhance the antimicrobial activity of *Artemisia argyi* against foodborne pathogens. Ten LAB strains were isolated from natural *Artemisia argyi* fermentation liquor (AAFL) and assessed for safety and antimicrobial ability. Therein, *Lactiplantibacillus plantarum* WLPL01, *Lacticaseibacillus casei* WLCA01, WLCA02, and WLCA03, and *Lactobacillus harbiness* WLHA01 were further evaluated for their potential probiotic properties (gastrointestinal tolerance and adhesion capacity). The results suggested that *L. plantarum* WLPL01 exhibited excellent properties and was, therefore, selected as the starter for *A. argyi* leaves fermentation. Then, *L. plantarum* WLPL01-fermented AAFL (AAFL-LP) was further investigated for its antimicrobial activity against foodborne pathogens. The results demonstrated that the inhibitory effect of AAFL-LP to foodborne pathogens, such as *Listeria monocytogenes* CMCC54007, *Salmonella* Typhimurium ATCC 13311, and *Candida albicans* ATCC 14053, was enhanced when compared to spontaneously fermented AAFL (AAFL-spontaneous). In addition, an analysis of the whole genome of *L. plantarum* WLPL01 revealed the presence of 13.9 kb long and 16 plantaricin-encoding loci (*pln* locus), and the increased antimicrobial activities of AAFL-LP might correlate with the production of bacteriocin. Our results indicate that *L. plantarum* WLPL01 can be used as a starter for *Artemisia argyi* fermentation to enhance its antimicrobial activity against foodborne pathogens.

**Keywords:** *Artemisia argyi*; *Lactiplantibacillus plantarum*; fermentation; antimicrobial activities; bacteriocin

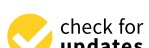



## 1. Introduction

*Artemisia argyi* (AA) is a herbaceous plant distributed in most regions of East Asia. It is frequently used as a traditional herbal medicine for the treatment of microbial infections, osteoarthritis, inflammatory diseases, asthma, diarrhea, hepatitis, malaria, cancer, and circulatory disorders [1,2] due to its abundant bioactive compounds such as *Artemisia* oil, glycosides, coumarins, flavonoids, polyacetylenes, monoterpenes, triterpenes, and sesquiterpene lactones [3–6]. Recently, AA found increasing applications as a feed additive in animal production for decreasing diarrhea and regulating gastrointestinal function [7], improving the breeding environment, and increasing animal feed intake [8]. Fermentation is a traditional process in which microorganisms or enzymes can generate biological compounds, and simultaneously the metabolism of microorganisms can be altered to produce new ingredients [9,10].

Traditional or artisanal fermented foods are deemed as abundant sources of lactic acid bacteria (LAB). For instance, *Lactiplantibacillus plantarum* Bom 816, *L. paracasei* B41, and *L. pentosus* N3 isolated from a traditional Turkish fermented cereal beverage boza showed

antimicrobial activity against *Escherichia coli*, *Klebsiella pneumoniae*, *Vibrio cholerae*, and *Bacillus subtilis* [11], and *L. plantarum* ULAG24 and ULAG11 from West African fermented cereals demonstrated antagonism to foodborne pathogens [12]. Gioia et al. reported that *L. plantarum* PCS20 was capable of effectively surviving in ground meat and performing antimicrobial activity in carnis against *Clostridium* [13].

On the other hand, LAB are widely used as a starter culture to control the fermentation rate, extend the shelf-life, improve functionality, ameliorate flavor, and ensure safety [14,15]. For example, the antagonistic effect of *A. princeps* was enhanced after fermentation with *Bifidobacterium infantis* K-525 [16]. Similarly, Jo's results confirmed that *Lactobacillus pentosus* SMB718 isolated from Korean traditional paste has many beneficial probiotic activities, such as antimicrobial and anti-inflammatory effects, and it could promote a significantly higher amount of allyl mercaptan production when used as a starter in garlic and onion fermentation [17].

However, reports that describe the distribution of native flora, e.g., the constitution of LAB strains in AAFL, are scarce, and research on the antagonistic effect of AAFL artificially fermented by native species of LAB is even more so. In this study, LAB strains from a naturally fermented *A. argyi* liquor were isolated and evaluated for the probiotic properties and food safety indexes. Based on the antibiotic sensitivity and survival ability in an artificial gastrointestinal tract (GI), a native strain of *L. plantarum* WLPL01 was screened out. Subsequently, *L. plantarum* WLPL01 was used as a starter to ferment *A. argyi*, and the antagonistic effect of *L. plantarum* WLPL01-fermented AAFL (AAFL-LP) against pathogens was compared with that of AAFL-spontaneous. Moreover, the bacteriocin-encoding genes in the genome of *L. plantarum* WLPL01 were bioinformatically analyzed. This work might provide a fundamental basis for using *A. argyi* fermentation liquor as a feed additive to control pathogens in the breeding industry.

## 2. Materials and Methods

### 2.1. Isolation and Identification of Strains from AAFL

*Artemisia argyi* fermentation liquor (AAFL) was presented by Professor Ziniu Yu (Huazhong Agricultural University, Wuhan, China). To isolate LAB strains, an aliquot of AAFL was 10-fold diluted and plated on De Man Rogosa and Sharpe (MRS) agar (Solarbio, Beijing, China), KFS agar (Hopebiol, Qingdao, China), and M17 agar (Hopebiol, Qingdao, China). After anaerobic incubation for 48 h at 37 °C in a thermostatic anaerobic incubator chamber (Gene Science, San Diego, CA, USA), single colonies were selected and purified on an MRS agar plate. Then, the obtained strains were screened by Gram staining and catalase testing. The Gram-positive and catalase-negative isolates were further identified by PCR with 16S rDNA universal primers, 27F and 1492R [18], the amplicons were sequenced by Sangon (Sangon, Shanghai, China), and the sequencing results were compared with the NCBI database using the BLAST algorithm (https://blast.ncbi.nlm.nih.gov/Blast.cgi, accessed on 1 October 2015) to identify species.

### 2.2. Phylogenetic Analysis

The genetic presence and comparison of 16S rRNA was performed by analyzing the nucleotide sequence data available at the National Center for Biotechnology Information (NCBI) database. The 16S rRNA gene sequences of related organisms were obtained from the NCBI database and compared with the sequences of our strains to establish the closest neighbor in the evolutionary tree. The neighbor-joining (NJ) phylogenetic tree with the p-distance model was constructed using the MEGA software (version 7.0, Mega Limited, Auckland, New Zealand).

### 2.3. Growth Curve and Acid Production

The strains were inoculated (1%, *v/v*) and incubated anaerobically at 37 °C, the $OD_{600}$ was measured using a microplate reader (Varioskan™ LUX, Waltham, MA, USA), and the pH was monitored using a pH meter (Bell, Dalian, China).

### 2.4. Antimicrobial Activity Assay

The antimicrobial activity of LAB was analyzed according to our previous report [19] with some modifications. Briefly, aliquots of 100 μL ($10^6$ CFU/mL) indicator strains viz. *Escherichia coli* O157:H7, *Staphylococcus aureus* CMCC26001, *Candida albicans* ATCC14053, *Listeria monocytogenes* CMCC54007, and *Salmonella* Typhimurium ATCC13311 were spread on the surface of LB agar plates, separately. Then, 200 μL of LAB supernatant was loaded into an Oxford cup (outer diameter 7.8 ± 0.1 mm, inner diameter 6.0 ± 0.1 mm, and height 10.0 ± 0.1 mm), which was placed on the surface of the agar. After incubation for 10 h at 37 °C, the inhibition zone around the cup was measured as a quantitative measure of inhibition ability. All the experiments were performed in triplicate.

### 2.5. Determination of Antibiotic Susceptibility

Antibiotic susceptibility was tested according to the method in [20] with slight modifications. Briefly, bacterial strains were inoculated (1%, *v/v*) in MRS broth supplemented with antibiotic (chloramphenicol, ampicillin, ciprofloxacin, polymyxin B, tetracycline, erythromycin, or gentamycin) (Solarbio, Beijing, China) at various final concentrations (2, 4, 8, 16, 32, 64, 128, 256, 512, and 1024 μg/mL) and the $OD_{600}$ was measured after a 24 h incubation period at 37 °C. All these experiments were performed in triplicate.

### 2.6. Detection of Biogenic Amine Production

The production of biogenic amine was tested according to the method in [21] with slight modifications. In brief, strains were incubated in the modified decarboxylase medium, which contained tryptone (0.5%, *w/v*), yeast extract (0.5%, *w/v*), NaCl (0.5%, *w/v*), glucose (0.1%, *w/v*), pyridoxal-5-phosphate (0.005%, *w/v*), Tween 80 (0.05%, *w/v*), $MgSO_4 \cdot 7H_2O$ (0.02%, *w/v*), $CaCO_3$ (0.01%, *w/v*), $MnSO_4 \cdot 4H_2O$ (0.005%, *w/v*), $FeSO_4 \cdot 7H_2O$ (0.004%, *w/v*), bacteriological agar (2%, *w/v*), and purple bromocresol (0.006%, *w/v*), in which the last one was used as a pH indicator. The precursor amino acids of each biogenic amine (histidine, lysine, ornithine, and tyrosine) (Solarbio, Beijing, China) were added individually to the culture medium at a final concentration of 2%, and pH was adjusted to 5.5 ± 0.1. After incubation, obvious pink circles around the growing single colonies were denoted as the positive result of biogenic amine.

### 2.7. Growth Characteristics of Strains in the Presence of Bile Salts and Acid

Bile salt resistance and acid tolerance assays were carried out according to the method in [22] with minor modifications. For the bile salt resistance assay, overnight cultures of the isolates were inoculated into MRS broth (1%, *v/v*) with 0.15%, 0.3%, and 0.45% ox bile salts (*w/v*) (Solarbio, Beijing, China). After incubation at 37 °C for 24 h, the $OD_{600}$ was measured. The results were expressed as the percentage of growth compared with the control (without ox bile salts). For the acid tolerance assay, bacterial cells were inoculated in MRS broth at pH 6.5, 4.5, and 3.5. Resistance was assessed in terms of colony counts and enumerated on an MRS agar plate after incubation at 37 °C for 0, 1, 3, and 12 h. All the assays were performed in triplicates.

### 2.8. Survival of Strains under Simulated Gastric and Intestinal Fluid

Overnight cultured cells were harvested by centrifugation (5000× *g*, 10 min, 4 °C), washed twice with PBS (pH 7.2), and resuspended in simulated gastric fluid (0.3% pepsin, 7 mmol/L KCl, 125 mmol/L NaCl, and 45 mmol/L $NaHCO_3$, pH 2.5) or simulated intestinal fluid (0.1 g/L pancreatin, 3 g/L ox bile salts, 0.835 g/L KCl, 6.5 g/L NaCl, 0.22 g/L $CaCl_2$, and 1.386 g/L $NaHCO_3$, pH 7.5) containing 10% (*w/v*) skimmed milk, and incubated for 120 min at 37 °C in anaerobic condition. Viable counts were counted on an MRS agar plate before and after the simulated gastric and intestinal fluid challenges. The tests were performed in triplicate.

### 2.9. Adhesive Ability of Strains to Caco-2 Cells

The adhesion of strains to Caco-2 cells was evaluated according to previous report [23]. Briefly, $5 \times 10^5$ Caco-2 cells were seeded in 6-well tissue culture plates and incubated at 37°C in a humidified 5% (*v/v*) $CO_2$ atmosphere for 24 h. One milliliter of each strain suspension ($1 \times 10^8$ CFU/mL in antibiotic-free DMEM) or DMEM solution (as control) was added into each well and incubated for 2 h at 37 °C. After incubation, the monolayers were washed to remove the unbound bacteria. The adherent bacteria were detached and dispersed in 1 mL of 0.05% Trypsin-EDTA, and then diluted serially and spread onto MRS agar plates for counting. The adhesion ability was expressed as the percentage of the adhesion of selected strains relative to the well-known probiotic strain *Lactobacillus rhamnosus* GG.

### 2.10. Preparation of AAFL-LP

*Artemisia argyi* was obtained from Qichun, Wuhan, and fermented using the traditional artisan method. Briefly, after drying and trimming, AA were cut and filtered, and then 5 g (10%, *w/v*) of sterile AA powders and 2.5 g (5%, *w/v*) of glucose were added to 50 mL of distilled water in a 100 mL conical flask (AA sterile). Then, *L. plantarum* WLPL01 ($5 \times 10^7$ CFU/mL) or 1 mL of 0.85% NaCl solution (for spontaneous fermentation as control) was inoculated. After mixing, samples were incubated at 37 °C for 5 days, and the obtained AAFL was named AAFL-LP or AAFL-spontaneous. The pH values and viable bacterial numbers of the fermented samples were monitored.

### 2.11. Antimicrobial Activity of AAFL-LP by Oxford Cup Assay

To investigate the antimicrobial activity, AAFL sterile, AAFL-spontaneous, and AAFL-LP were extracted by ultrasonic extraction at 25 °C for 1 h with 80% (*v/v*) ethanol. After filtering through filter paper, the extracts were vacuum concentrated and freeze-dried. The antimicrobial activity of crude extracts was tested in an agar spot assay, as described above.

### 2.12. Genomic DNA Extraction and Genome Analysis

The genomic DNA of *L. plantarum* WLPL01 was extracted using the cetyltrimethyl ammonium bromide (CTAB) method, and the DNA concentration, quality, and integrity were determined using a Qubit Flurometer (Invitrogen, Waltham, MA, USA) and a NanoDrop Spectrophotometer (Thermo Scientific, Waltham, MA, USA). Sequencing libraries were generated using the TruSeq DNA Sample Preparation Kit (Illumina, San Diego, CA, USA) and the Template Prep Kit (Pacific Biosciences, Menlo Park, CA, USA). The genome sequencing was then performed by the Personal Biotechnology Company (Shanghai, China) using the Pacific Biosciences platform and the Illumina Miseq platform.

Data assembly proceeded after adapter contamination removal and data filtering using AdapterRemoval [24] and SOAPec [25]. The filtered reads were assembled using SPAdes [26] and A5-miseq [27] to construct scaffolds and contigs. The Canu [28] software was used to assemble the data obtained by Pacbio platform sequencing. Subsequently, all the assembled results were integrated to generate a complete sequence. Finally, the genome sequence was acquired after the rectification using the pilon software [29].

### 2.13. Identification of Bacteriocin-Encoding Genes of L. plantarum WLPL01

Different plantaricin genes were identified in whole-genome sequence of *L. plantarum* WLPL01 with a similarity search of the sequence using BLASTP and compared with known plantaricin genes [30]. Subsequently, the organization of plantaricin genes of the individual strain of *L. plantarum* was analyzed using the bacteriocin database BAGEL4 [31].

### 2.14. Statistical Analysis

The results were evaluated using analysis of variance (ANOVA) and Tukey test; *p*-values less than 0.05 were considered statistically significant. The analysis was performed with GraphPad Prism version 7.0 (GraphPad Software, Inc., La Jolla, CA, USA).

## 3. Results

### 3.1. Isolation and Identification of Strains from AAFL

Ten isolates of LAB strains from *A. argyi* fermentation liquid (AAFL) were identified as one strain of *Lactiplantibacillus plantarum* (WLPL01), three strains of *Lacticaseibacillus casei* (WLCA01, WLCA02, and WLCA03), one strain of *Lactobacillus harbiness* (WLHA01), and five strains of *Lentilallactobacillus buchneri* (WLBU01, WLBU02, WLBU03, WLBU04, and WLBU05). The relevant phylogenetic tree of these strains is shown in Figure 1.

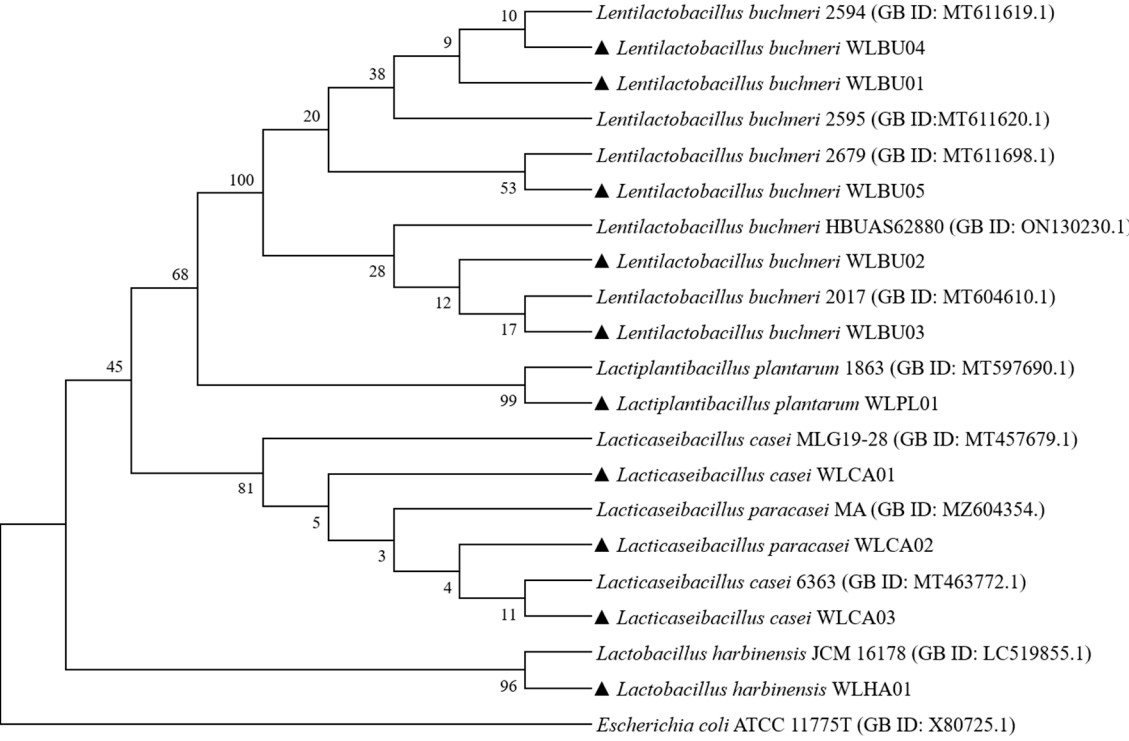

**Figure 1.** Phylogenetic tree of the LAB isolates from *Artemisia argyi* fermentation liquor (AAFL) based on 16S rRNA gene sequences. *E. coli* ATCC 11775T was taken as an out-group. Bootstrap values are given at branching points. Filled upward triangles represent the LAB strains isolated in this study.

### 3.2. Growth Curve and Acid Production of Strains

The growth curve and pH value of the 10 isolates are shown in Figure 2. Strains were categorized into two groups: one group included one of *L. plantarum* strain, three of *L. casei* strains, and one of *L. harbinesis* strain, with strains entering the stationary phase at 12 h, with a final pH of 3.8; the other group included five *L. buchneri* strains that had a long logarithm growth phase (48 h) and final higher pH of 4.3. Among these 10 strains, *L. plantarum* WLPL01 showed the highest growth tendency with rapid acidification.

### 3.3. Antibiotic Sensitivity and Biogenic Amine Test of Strains

Antibiotic resistance is a risk factor in food safety. We chose typical antibiotics, including cell wall synthesis inhibitors (ampicillin), DNA synthesis inhibitors (ciprofloxacin), and protein synthesis inhibitors (tetracycline, gentamycin, erythromycin, and chloramphenicol), to evaluate the antibiotic sensitivity of 10 strains, compared with the breakpoints set by EUCAST (2017). The results showed that one *L. harbinesis* (WLHA01) and two out of three *L. casei* strains (WLCA01 and WLCA02) were found to be resistant to gentamycin, four out of five *L. buchneri* (WLBU02, WLBU03, WLBU04, and WLBU05) strains showed resistance to tetracycline, and *L. plantarum* WLPL01, *L. casei* WLCA03, and *L. buchneri* WLBU01 showed susceptibility to all the tested antibiotics (Table 1).

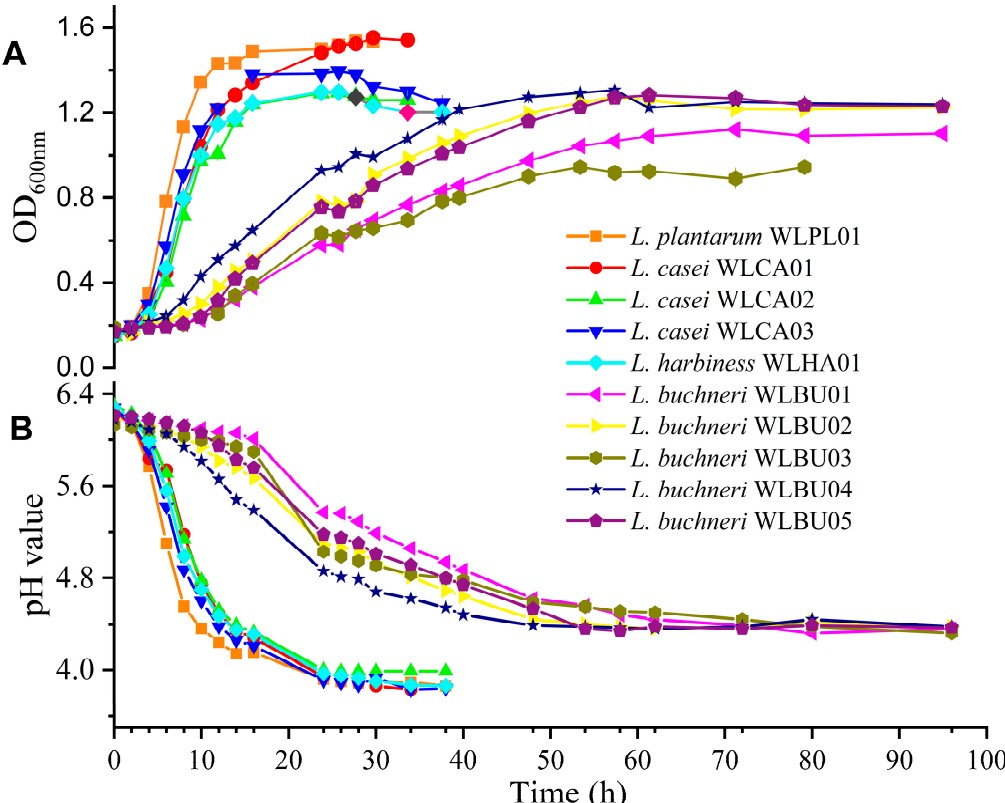

**Figure 2.** The growth curves (**A**) and pH value (**B**) of the LAB strains from AAFL under anaerobic conditions in MRS broth at 37 °C.

**Table 1.** MIC distribution of LAB strains isolated from AAFL to antibiotics.

| Strains | MICs [a] (μg/mL) | | | | | | |
|---------|-----|-----|-----|------|-----|-----|-----|
| | **CHL** | **AMP** | **CIP** | **POLY** | **TET** | **ERY** | **GEN** |
| WLPL01 | 2 | 2 | 64 | 128 | 16 | ≤1 | 64 |
| WLCA01 | 2 | 2 | 64 | 512 | 1 | ≤1 | 128 [R] |
| WLCA02 | 2 | 2 | 128 | 512 | 1 | ≤1 | 128 [R] |
| WLCA03 | ≤1 | 2 | 128 | 256 | ≤1 | ≤1 | 32 |
| WLHA01 | 2 | 2 | 64 | 256 | 2 | ≤1 | 128 [R] |
| WLBU01 | 4 | 2 | 64 | 4 | 16 | ≤1 | ≤1 |
| WLBU02 | 8 | 2 | 128 | 4 | 32 [R] | ≤1 | ≤1 |
| WLBU03 | 4 | 2 | 128 | 4 | 32 [R] | ≤1 | ≤1 |
| WLBU04 | 4 | 2 | 128 | 8 | 32 [R] | ≤1 | ≤1 |
| WLBU05 | <1 | 2 | 128 | 4 | 32 [R] | ≤1 | ≤1 |

[a] MIC: minimum inhibitory concentration; CHL: chloramphenicol; AMP: ampicillin; CIP: ciprofloxacin; POLY: polymyxin B; TET: tetracycline; ERY: erythromycin; GEN: gentamycin. R: resistance according to the European Committee on clinical breakpoints for bacterial (http://www.eucast.org/clinical_breakpoints/, accessed on 25 May 2017) and antimicrobial susceptibility testing (http://www.eucast.org/mic_distributions_and_ecoffs/, accessed on 25 May 2017) in 2017.

Free from or reduction in biogenic amines is a criterion for screening LAB strains. The result showed that strains of *L. buchneri,* i.e., WLBU03 and WLBU05, were positive in the yielding of putrescine and cadaverine, respectively, while other strains were all negative in biogenic amine production.

### 3.4. Antimicrobial Activity of Strains

The antimicrobial spectrum of 10 strains was investigated using five kinds of pathogens as indicators. As shown in Figure 3, *L. plantarum* WLPL01, *L. harbinesis* WLHA01, and

*L. casei* (WLCA01, WLCA02, and WLCA03) showed a broad spectrum against Gram-positive (*L. monocytogenes* and *S. aureus*), Gram-negative (*E. coli* and *S*. Typhimurium), and fungus (*C. albicans*). From these, *L. plantarum* demonstrated the strongest antagonistic effect. In contrast, all the five strains of *L. buchneri* showed no antimicrobial activity except for *E. coli* O157:H7 and *S*. Typhimurium (with a very small inhibition zone for WLBU01 and WLBU02).

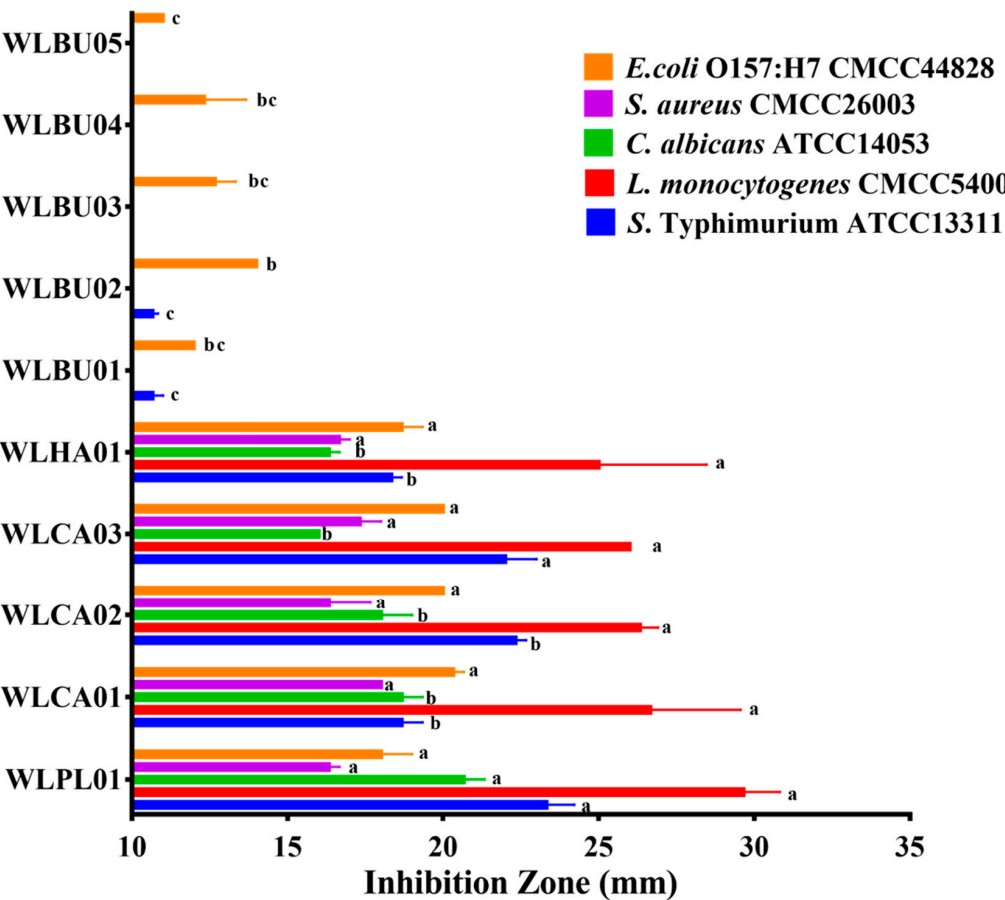

**Figure 3.** Antagonistic activities of the LAB strains from AAFL against *Escherichia coli* O157:H7, *Staphylococcus aureus* CMCC26001, *Candida albicans* ATCC14053, *Listeria monocytogenes* CMCC54007, and *Salmonella* Typhimurium ATCC13311 in well diffusion assay. Values are presented as means ± standard deviation in triplicate. Different letters represent significant differences ($p < 0.05$).

*3.5. Survival of Strains in Mimic GI and Adhesion to Caco-2 Cells*

As probiotics, lactic acid bacteria are expected to colonize in the gut under the stress of GI condition. Therefore, based on the results of growth curves, antimicrobial properties, and biogenic amine production, five strains (*L. plantarum* WLPL01, *L. casei* WLCA01, WLCA02, WLCA03, and *L. harbinesis* WLHA01) were selected for further testing in an artificial GI.

In the bile salt tolerance test (Table 2), *L. plantarum* WLPL01, *L. casei* WLCA01, WLCA02, WLCA03, and *L. harbinesis* WLHA01 all showed a slight and nonstatistically significant decrease in survival rates, in a range of 0.15–0.45% bile salts. For low pH tolerance (Figure 4A), all five strains maintained values as high as $10^7$ CFU/mL, even under pH 3.5 for 12 h, showing a significant growth tendency in MRS medium at pH 4.5 for 12 h ($p < 0.05$). To further evaluate the survival capability of the five strains, separate simulated gastric digestion and intestinal digestion tests were performed. As shown in Figure 4B, WLPL01 and WLHA01 maintained the initial level of viable counts, while the counts of all three *L. casei* strains increased significantly ($p < 0.01$) in the simulated gastric fluid for 2 h. However, *L. casei* WLCA01 and WLCA02 were significantly ($p < 0.001$) decreased after incubation

with simulated intestinal fluids for 2 h, while WLPL01 and WLHA01 were still stable in the simulated intestinal fluids.

**Table 2.** Tolerance of five selected LAB strains to bile salt stress. Values are presented as means ± standard deviation in triplicate.

| Strain | Resistance to Bile Salt (%) | | |
|---|---|---|---|
| | 0.15% | 0.30% | 0.45% |
| WLPL01 | 11.90 ± 2.04 | 11.23 ± 2.15 | 12.01 ± 4.47 |
| WLCA01 | 14.43 ± 0.17 | 13.39 ± 0.42 | 12.50 ± 0.87 |
| WLCA02 | 14.50 ± 0.44 | 13.04 ± 0.42 | 11.81 ± 0.39 |
| WLCA03 | 13.35 ± 0.97 | 12.69 ± 0.62 | 11.73 ± 0.81 |
| WLHA01 | 10.34 ± 0.10 | 10.23 ± 0.17 | 9.56 ± 0.17 |

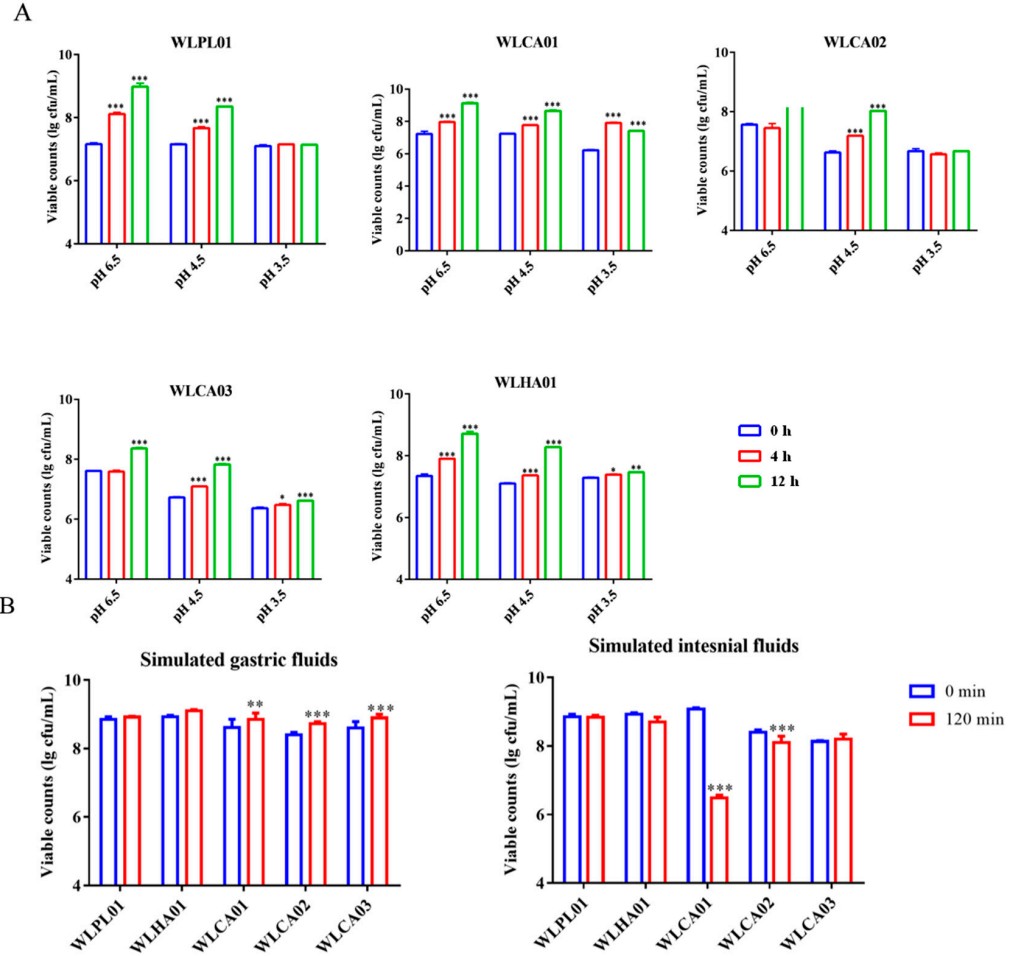

**Figure 4.** Low pH tolerance (**A**) and survival capacity in simulated gastric and intestinal fluids (**B**) of the five selected LAB strains. Values are presented as means ± standard deviation in triplicate. Significant differences compared with the respective control (0 h or 0 min) are denoted as * $p < 0.05$; ** $p < 0.01$; *** $p < 0.001$.

Adherence on epithelial cells is considered to be an important and initial event for lactic acid bacteria to function as probiotics. *Lactobacillus rhamnosus* GG (LGG), as an outstanding *Lactobacillus* due to its excellent probiotic function, is often selected as a positive control when exploring other probiotic functions. As shown in Figure 5, the highest adhesion ratio of 82.17 ± 6.85% was obtained for WLCA03, followed by WLPL01 and WLCA01. All of them showed a weaker adhesion ability when compared with the strain of LGG, which was also used as a positive control. Its adhesion ability (%) was set as 100%.

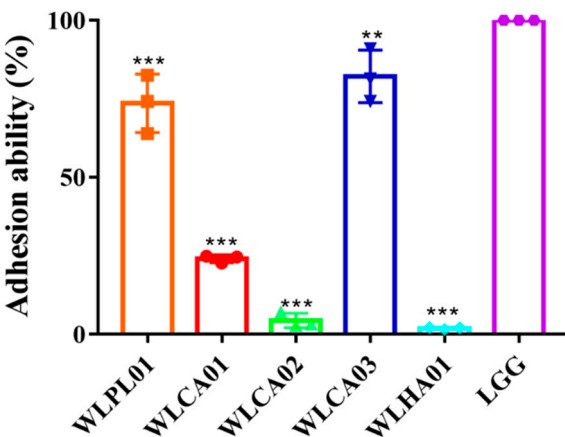

**Figure 5.** The adhesion of the five selected LAB strains to Caco-2 cells. The adhesion ability was expressed as a percent compared to the LGG group. Values are presented as means ± standard deviation in triplicate. Significant differences compared with LGG group are denoted as ** $p < 0.01$; *** $p < 0.001$.

### 3.6. Antimicrobial Activity of L. plantarum WLPL01-Fermented AAFL (AAFL-LP) against Pathogens

Chemical and microbiological changes were monitored during the fermentation of *A. argyi* by *L. plantarum* WLPL01 (Figure 6A). Generally, viable counts of WLPL01 achieved the highest value of $8.51 \pm 0.09$ log CFU/mL at 48 h, while spontaneous fermentation took 96 h to reach a peak of $8.56 \pm 0.03$ log CFU/mL, under the condition of *A. argyi* supplied with an extra carbon source. For the pH, inoculation of WLPL01 resulted in a rapid decease in the pH value (final pH 3.3). All the above results suggested that fermented *A. argyi* might be affected by the existence of either WLPL01 or native flora of *A. argyi*. As shown in Figure 6B, neither AAFL-LP nor AAFL-spontaneous exhibited a significantly enhanced antagonistic ability against all of the tested pathogens. For *C. albicans*, both AAFL-LP and AAFL-spontaneous showed enhanced antagonistic ability; therein, AAFL-spontaneous at day 5 and AAFL-LP at day 1, 3, and 5 significantly increased the inhibition zone compared to unfermented *A. argyi* liquor ($p < 0.05$). Remarkably, AAFL-LP significantly strengthened the inhibition ability at day 3, compared with AAFL-spontaneous ($p < 0.05$). For *L. monocytogenes* and *S.* Typhimurium, AAFL-LP and AAFL-spontaneous also achieved a significant inhibition. Compared with AAFL-spontaneous, AAFL-LP showed a significantly higher inhibition ability against *L. monocytogenes* at day 3 ($p < 0.001$) and day 5 ($p < 0.01$), and *S.* Typhimurium exhibited a higher ability at day 5 ($p < 0.05$).

We further used *S.* Typhimurium as an example to test the inhibitory effect of AAFL-LP with different concentrations. As shown in Figure 7, the viable counts of *S.* Typhimurium were significantly decreased ($p < 0.001$) when co-cultured with 25 mg/mL AAFL-LP, and no living cell was observed when co-cultured with 50 mg/mL of AAFL-LP. Moreover, the morphology of *S.* Typhimurium observed by SEM showed that *S.* Typhimurium cells were rough, severely wrinkled, and collapsed when co-cultured with 50 mg/mL of AAFL-LP.

### 3.7. Genome Analysis and Bacteriocin Identification of L. plantarum WLPL01

An analysis of the whole-genome sequence for the *L. plantarum* WLPL01 (NZ_CP122974.1) was carried out. The result (Figure 8A) showed that the genome of *L. plantarum* WLPL01 contains a circular chromosome, with a size of 3,142,471 bp, an average G + C content of 44.68%, containing 2946 candidate protein-coding genes (with an average size of 840 bp), among which 2942 proteins were found to be functionally categorized. The chromosomal properties of *L. plantarum* WLPL01 are summarized in Table 3, with *L. plantarum* WCFS1 (GenBank Accession no. AL935263) as a reference. In addition, the chromosome of *L. plantarum* WLPL01 has 147 RNA genes including 71 tRNA, 16 rRNA, and 60 ncRNA. In addition, the genome of *L. plantarum* WLPL01 has four plasmids, with sizes of 42,746,

38,441, 10,743, and 6349 bp, respectively, and a G + C content that ranged from 35.85% to 40.77%.

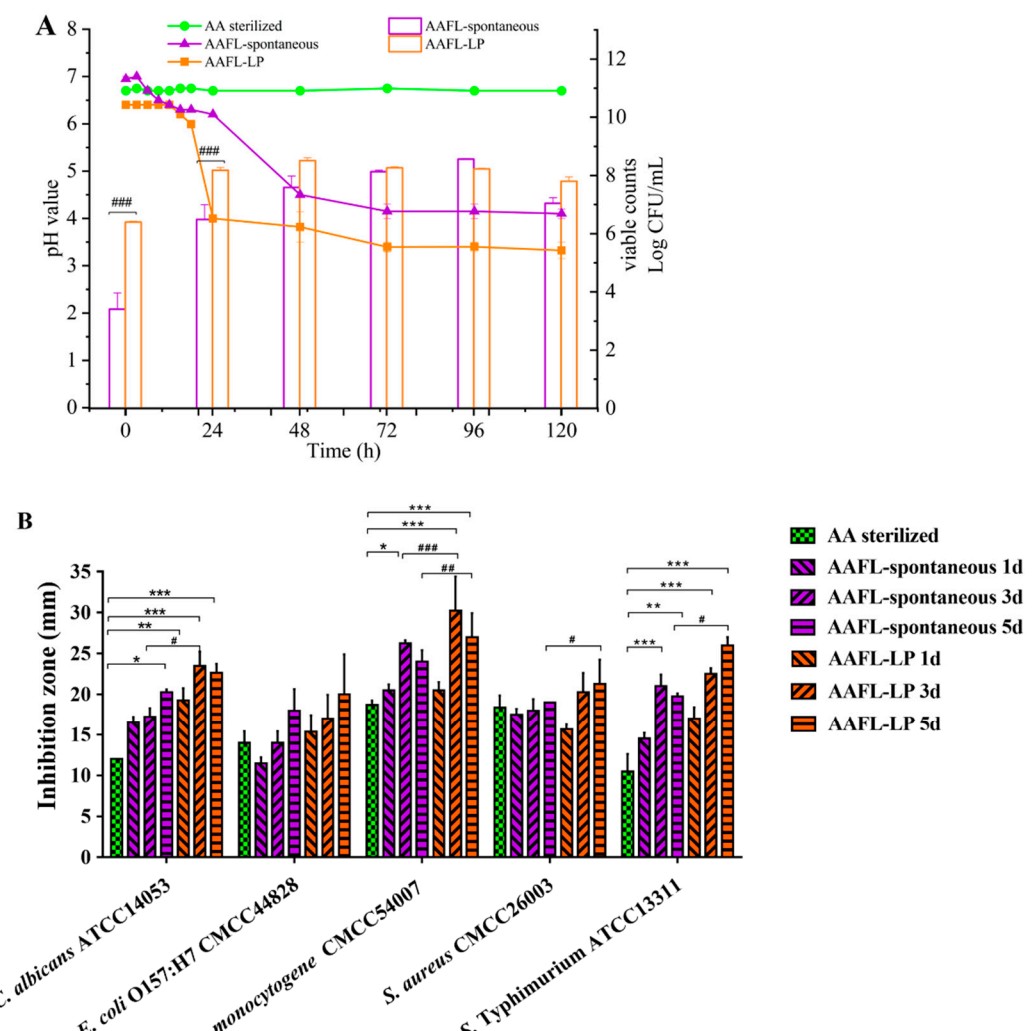

**Figure 6.** Effect of fermentation on viable counts of microbial cells and pH (**A**), antimicrobial activity (**B**) of AAFL. Values are presented as means ± standard deviation in triplicate. * Denotes significant differences compared with control (AA sterilized) (* $p < 0.05$; ** $p < 0.01$, *** $p < 0.001$); [#] Denotes significant difference between AAFL-spontaneous and AAFL-LP ([#] $p < 0.05$, [##] $p < 0.01$, [###] $p < 0.001$).

The capacity of AAFL-LP against *S.* Typhimurium might correlate with the bacteriocin produced by *L. plantarum* WLPL01 during fermentation; thus, we identified the bacteriocin-encoding genes in the *L. plantarum* WLPL01 genome. Screening of the entire genome of *L. plantarum* WLPL01 revealed that the bacteriocin-encoding locus (*pln* locus) was located in a 13.9 kb long region organized in an operon-like structure, which consists of 16 genes (*pln*), including *pln* genes of *pln*J, *pln*K, *pln*M, *pln*N, *pln*O, *pln*Q, *pln*B, *pln*D, *pln*L, *pln*E, *pln*F, *pln*H, and *pln*T, one peptide export ABC transporter, and one transcriptional regulator (Figure 8B).

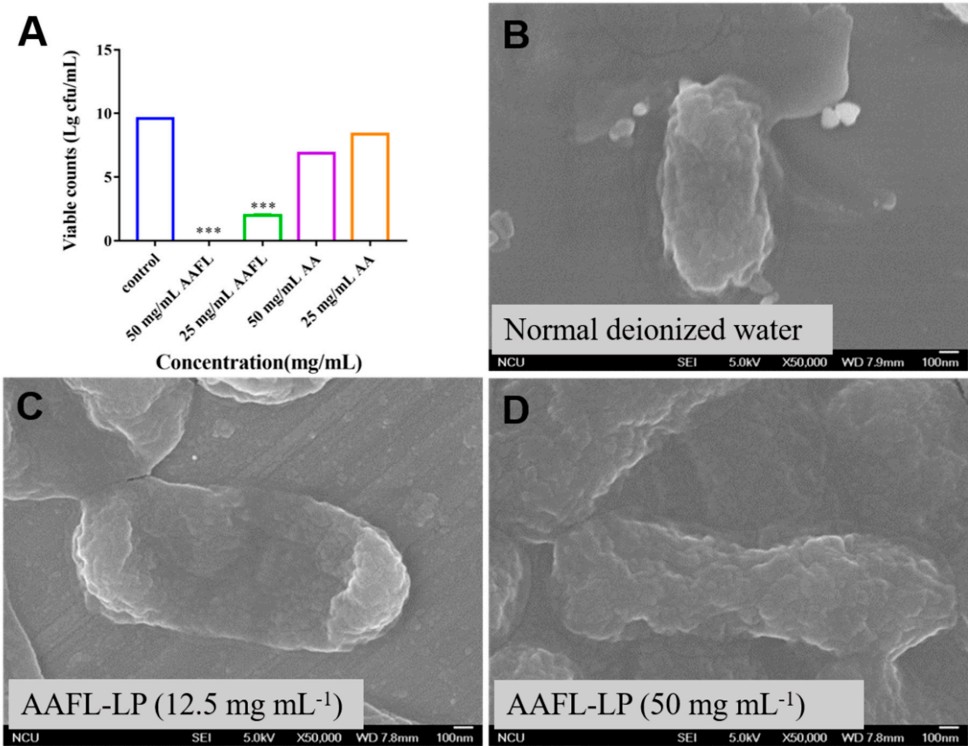

**Figure 7.** In vitro antimicrobial activity of AAFL-LP against *S.* Typhimurium ATCC 13311 (**A**) and scanning electron micrographs of *S.* Typhimurium after 2 h in presence of normal deionized water (**B**), AAFL-LP (12.5 mg/mL) (**C**), and AAFL-LP (50 mg/mL) (**D**). * Denotes significant differences compared with control (*** $p < 0.001$).

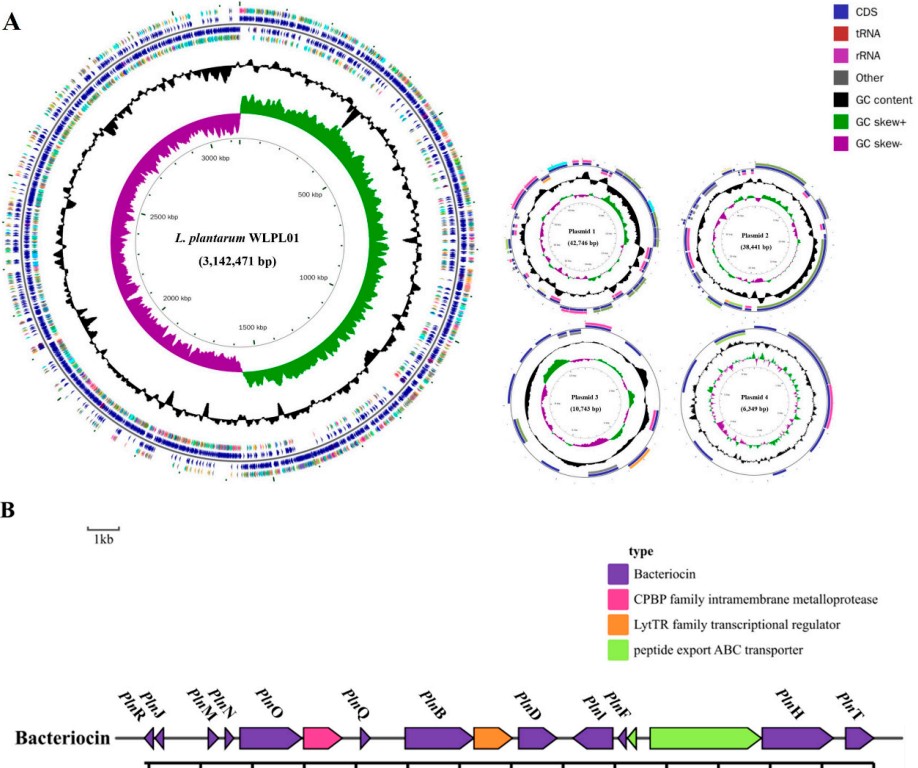

**Figure 8.** Circular map of the *L. plntarum* WLPL01 genome (**A**) and genetic organization map of the plantaricins gene cluster of *L. plantarum* WLPL01 (**B**).

**Table 3.** Comparison of the features of the *L. plantarum* WLPL01 genome with the reference genome.

| Strain | *L. plantarum* WLPL01 | *L. plantarum* WCFS1 |
|---|---|---|
| Source | *A. argyi* fermentation liquor | Human saliva |
| Chromosomal (bp) | 3,142,471 | 3,308,274 |
| G + C Content (%) | 44.68% | 45.6% |
| Total number of genes | 2946 | 3174 |
| Coding genes | 2942 | 3063 |
| Total No. of RNA | 146 | 88 |
| No. of tRNA | 70 | 70 |
| No. of rRNA | 16 | 15 |
| No. of nc RNA | 60 | 3 |
| No. of CRISPER array | 0 | 0 |
| No. of prophage region | 8 | 4 |

## 4. Discussion

Considerable functional studies have been conducted on *A. argyi* to discover its physiochemical properties. These have been focused on its essential oil, flavone, and polysaccharide. Little is known about the function of *A. argyi* after fermentation or its composition and contribution to the microbial inside. In this study, according to its primary properties, the *L. plantarum* WLPL01 strain was screened out from 10 indigenous strains in AAFL, and its safety and potential antimicrobial capacity was tested for the fermentation of *A. argyi*, targeting its use as a starter culture and as a feed additive.

It has been suggested that the low diversity of LAB in grass is possibly attributed to the low water solvent carbohydrate levels in it [1]. In our work, 10 strains belonged to limited species of *L. casei*, *L. harbinesis*, *L. plantarum,* and *L. buchneri* were isolated. This is consistent with the finding of Mundt and Hammer (1968) [32], in which only two strains of *L. plantarum*, two of *L. brevis,* and one strain of *L. buchneri* were isolated in greens.

The safety evaluation of the selected LAB strains was a priority to enable consumer consumption [33], which included the potential to produce biogenic amines, antibiotic resistance, toxin genes, and hemolysis potential. For antibiotic resistance, Mathur and Singh proposed that, when used as a starter culture, the risk of the undesirable transfer of resistance from LAB to the host or conferment of resistance to endogenous bacteria should be avoided [34]. In this aspect, strains of *L. plantarum* WLPL01, *L. casei* WLCA03, and *L. buchneri* WLBU01 exhibited sensitivity to chloramphenicol, ampicillin, ciprofloxacin, tetracycline, erythromycin, and gentamycin, suggesting their advantage as a starter culture. In addition, the production of biogenic amine during fermentation might potentiate toxic effects; therefore, free or reduced biogenic amines were critical for the selection of potential LAB strains. We found that two of the five *L. buchneri* strains, i.e., WLBU03 and WLBU05, were positive for putrescine and cadaverine production, respectively. Similarly, W. Straub reported that eight out of eleven *L. buchneri* strains produced histamine in a synthetic media [35]; another study also reported that *L. buchneri* produced extraordinary amounts of histamine in cheese, even in very low numbers in the inoculum [36].

The biomass and acidification were important criteria in screening strains in the starter culture. It was proposed that the fast-acidifying strains are good candidates in the dairy fermentation process as primary starter organisms [37]. Our results showed that five strains: *L. plantarum* WLPL01, *L. casei* WLCA01, WLCA02, WLCA03, and *L. harbinesis* WLHA01, had a shorter logarithm time and faster decline in pH than the others: *L. buchneri* WLBU01, WLBU02, WLBU03, WLBU04, and WLBU05.

Considering the results of the growth rate, the basic safety evaluation, and the microbial inhibition test, five strains (WLPL01, WLCA01, WLCA02, WLCA03, and WLHA01) were further selected to test their abilities via a survival and adhesion in vitro assay. All strains showed a similar bile salt resistance ability, and an increased tendency for viable cell counts in pH 4.5 for 12 h, and some of them maintained a slight increase even in pH 3.5. In the simulated gastric and intestinal test, all five strains, except for WLCA01, maintained

values as high as $10^8$ CFU/mL for 2 h. Besides the basic criteria for LAB isolation, more standards need to be met for the isolation and intended use as a starter culture.

Based on the results above, *L. plantarum* WLPL01 was chosen as a starter culture for the fermentation of *A. argyi* and to further analyze the antimicrobial activates of AAFL-LP. Neither AAFL-LP nor AAFL-spontaneous maintained cell counts as high as $10^7$–$10^8$ CFU/mL after fermentation for 5 days, with AAFL-LP decreasing in time to reach stability and maintaining a lower pH of 3.5 as compared with AAFL-spontaneous. Of the five tested pathogens, AAFL-LP significantly enhanced the antimicrobial activities of *A. argyi* against *C. albicans* and its antagonistic potential was even superior to that of AAFL-spontaneous. In the former inhibition assay of 10 LAB strains, WLPL01 also showed the best antagonistic effect against *C. albicans*. Similarly, the higher antimicrobial activities of AAFL-LP for *L. monocytogenes* and *S.* Typhimurium were achieved in day 5. Our findings indicated that LAB fermentation might strengthen the antagonistic abilities of *A. argyi* against certain pathogens and the activities might change over time. Previous studies reported the antagonistic effect of the essential oil of *Artemisia* species to *C. albicans* and *L. monocytogenes* [38] or the inhibitory effects of *L. plantarum* strains against *C. albicans* [39] and *L. monocytogenes* [40,41]. The enhanced antimicrobial activities of *L. plantarum*-fermented *A. argyi* against *L. monocytogenes* and *C. albicans* were verified in this study. In addition, *Salmonella* is the most common foodborne pathogen, affecting millions of people annually, sometimes with severe and fatal outcomes [42]. Examples of foods involved in outbreaks of *Salmonella* include eggs, poultry, and other products of animal origin (https://www.who.int/news-room/fact-sheets/detail/food-safety, accessed on 1 January 2020). In this work, we found that AAFL-LP strengthened the inhibitory ability against *S.* Typhimurium compared to sterile AA and AAFL-spontaneous.

Bacteriocins are antibacterial peptides produced by bacteria. They have antibacterial properties and can inhibit the growth of different microorganisms. Because of their antibacterial activity, they have attracted much interest in the food industry as natural preservatives. *L. plantarum* WLPL01 was identified to encode genes for two peptides, plantaricin *pln*JK (class IIb) and *pln*EF (class I), previously described for other *L. plantarum* strains [43]. The presence of two *pln* loci-encoding plantaricins from two different classes contributes to the broad inhibitory spectrum of *L. plantarum* [44]. Moreover, the *pln* locus was also found to contain *pln*N described in other *L. plantarum* strains, such as C11, WCFS1, and V90. The presence of plantaricin secretary genes *pln*H has also been confirmed in *L. plantarum* strains DHCU70 and DKP1, which are involved in the ABC transport system [45].

## 5. Conclusions

In conclusion, we found that AAFL-LP significantly enhanced the antimicrobial activities, effectively preventing *S.* Typhimurium in vitro, which might correlate with the bacteriocins produced by *L. plantrum* WLPL01 during fermentation. Hence, our work might provide a fundamental basis for using *L. plantrum* WLPL01 and AAFL-LP as feed additives to control pathogens in the breeding industry.

**Author Contributions:** Conceptualization, X.T. and H.W.; Methodology, H.Z.; Software, H.Z. and Y.H.; Validation, Q.W., Q.L. and L.H.; Data Curation, H.Z. and Y.H.; Writing—Original Draft Preparation, H.Z. and Y.H.; Writing—Review & Editing, X.T. and H.W.; Project Administration, X.T. and H.W. All authors have read and agreed to the published version of the manuscript.

**Funding:** This work was supported by National Natural Science Foundation of China (32060030).

**Institutional Review Board Statement:** Not applicable.

**Informed Consent Statement:** Not applicable.

**Data Availability Statement:** Not applicable.

**Conflicts of Interest:** The authors declare no conflict of interest.

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
