# Peer review of "Evaluation of Probiotic Strains Isolated from Artemisia argyi Fermentation Liquor and the Antagonistic Effect of Lactiplantibacillus plantarum against Pathogens"

_fermentation, doi:10.3390/fermentation9060536_

Round 1

Reviewer 1 Report

The Ms (Evaluation of probiotic strains isolated from Artemisia argyi fermentation liquor and the antagonistic effect of Lactiplantibacillus plantarum against pathogens )  is very good work . well designed, presented and discussed

However some minor points should be addressed

1-   L80 delete  respectively

2-  L104  replace respectively by separately

3-  L143 correct g/L1

4-  Table 2 calrify the unit of vales used in this table as survival rate is not clear

5-  Fig 5 clarify the LGG in more detail

Author Response

Review#1: The Ms (Evaluation of probiotic strains isolated from Artemisia argyi fermentation liquor and the antagonistic effect of Lactiplantibacillus plantarum against pathogens) is very good work. well designed, presented and discussed. However some minor points should be addressed.

Point 1: L80 delete respectively

Response 1: Thanks for your kind advice. We have deleted the “respectively” as you suggested (Line 80).

Point 2: L104 replace respectively by separately

Response 2: Thanks for your valuable advice. We have replaced “respectively” by “separately” as you kindly suggested (Line 104).

Point 3: L143 correct g/L1

Response 3: Thanks for your constructive suggestion. We have corrected “g/L1” as you kindly suggested (Line 143).

Point 4: Table 2 calrify the unit of vales used in this table as survival rate is not clear

Response 4: Thanks for your constructive suggestion. We detected bile salts resistance by measuring optical density at 600 nm (OD600) of incubations at 37°C for 24 h of the isolates and MRS broth (1%, v/v) with 0.15%, 0.3%, and 0.45% ox bile salts (w/v). As you suggested, though the survival rate after incubation with different bile acid salts clarified in Table 2 revealed some information, the survival rate was not totally clarified cause the specific relationship between the OD600 and the viable counts was not legible. Therefore, we will definitely consider and apply the comment to our subsequent work exploration in future.

Point 5: Fig 5 clarify the LGG in more detail

Response 5: Thanks for your constructive suggestion. We have clarified the LGG in a more detailed way and now the sentence turned into “Lactobacillus rhamnosus GG (LGG), as a star Lactobacillus due to its excellent probiotic function, is often selected as a positive control when exploring other probiotic functions. As shown in Figure 5, the highest adhesion ratio of 82.17 ± 6.85% was obtained in WLCA03, followed by WLPL01 and WLCA01. All of them showed weaker adhesion ability when compared with strain of LGG, which is also used as a positive control and the adhesion ability (%) of it was set as 100%.” (Line 278-283).

Reviewer 2 Report

The study may result of interest for the journal Fermentation. The following minor points are recommended for improvement and publication:

- Please, report L. harbiness with the correct nomenclature within the text.

- No statistical analysis was performed on data reported in Fig 2. How Authors can establish that “L. plantarum WLPL01 showed the highest growth tendency with rapid acidification” (L 213-214)?

Author Response

Review#2: The study may result of interest for the journal Fermentation. The following minor points are recommended for improvement and publication:

Point 1: Please, report L. harbiness with the correct nomenclature within the text.

Response 1: Thanks for your kind advice. We have corrected “L. harbiness” into the correct nomenclature “L. harbinensis” within the text and marked them into red in the revised manuscript.

Point 2: No statistical analysis was performed on data reported in Fig 2. How Authors can establish that “L. plantarum WLPL01 showed the highest growth tendency with rapid acidification” (L 213-214)?

Response 2: Thanks for your kind advice. We draw this conclusion based on some mathematical knowledge. As shown in Figure 2, whether growth curve or acid production curve of L. plantarum WLPL01 had the maximum slopes, which help us preliminarily concluded that it “L. plantarum WLPL01 showed the highest growth tendency with rapid acidification”.